# Fast and Cost-effective Speculative Edge-Cloud Decoding with Early Exits

**Yeshwanth Venkatesha**                                     *yeshwanth.venkatesha@yale.edu*
*Department of Electrical Engineering*
*Yale University*
*New Haven, CT, USA*

**Souvik Kundu**                                             *souvikk.kundu@intel.com*
*Intel Labs*
*San Diego, CA, USA*

**Priyadarshini Panda**                                      *priya.panda@yale.edu*
*Department of Electrical Engineering*
*Yale University*
*New Haven, CT, USA*

**Reviewed on OpenReview:** `https://openreview.net/forum?id=PTIUjARnbc`

## Abstract

Large Language Models (LLMs) enable various applications on edge devices such as smartphones, wearables, and embodied robots. However, their deployment often depends on expensive cloud-based APIs, creating high operational costs, which limit access for smaller organizations and raise sustainability concerns. Certain LLMs can be deployed on-device, offering a cost-effective solution with reduced latency and improved privacy. Yet, limited computing resources constrain the size and accuracy of models that can be deployed, necessitating a collaborative design between edge and cloud. We propose a fast and cost-effective speculative edge-cloud decoding framework with a large target model on the server and a small draft model on the device. By introducing early exits in the target model, tokens are generated mid-verification, allowing the client to preemptively draft subsequent tokens before final verification, thus utilizing idle time and enhancing parallelism between edge and cloud. Using an NVIDIA Jetson Nano (client) and an A100 GPU (server) with Vicuna-68M (draft) and Llama2-7B (target) models, our method achieves up to a 35% reduction in latency compared to cloud-based autoregressive decoding, with an additional 11% improvement from preemptive drafting. To demonstrate real-world applicability, we deploy our method on the Unitree Go2 quadruped robot using Vision-Language Model (VLM) based control, achieving a 21% speedup over traditional cloud-based autoregressive decoding. These results demonstrate the potential of our framework for real-time LLM and VLM applications on resource-constrained edge devices.

## 1 Introduction

Large Language Models (LLMs) have become pivotal in advancing artificial intelligence, transforming natural language processing (NLP), and enabling a wide range of applications such as chatbots, virtual assistants, robotics, translation, coding, and content generation Zeng et al. (2023); Huang et al. (2024); Sun et al. (2024); Zhang et al. (2023). Their importance lies in their ability to understand and generate human-like text, making interactions between humans and machines seamless and suggesting potential emergent capabilities Wei et al. (2022). Recent advances include large-scale models like OpenAI's GPT Radford et al. (2019); Brown (2020); Achiam et al. (2023), Meta's LLaMA Touvron et al. (2023); Dubey et al. (2024), and Google's

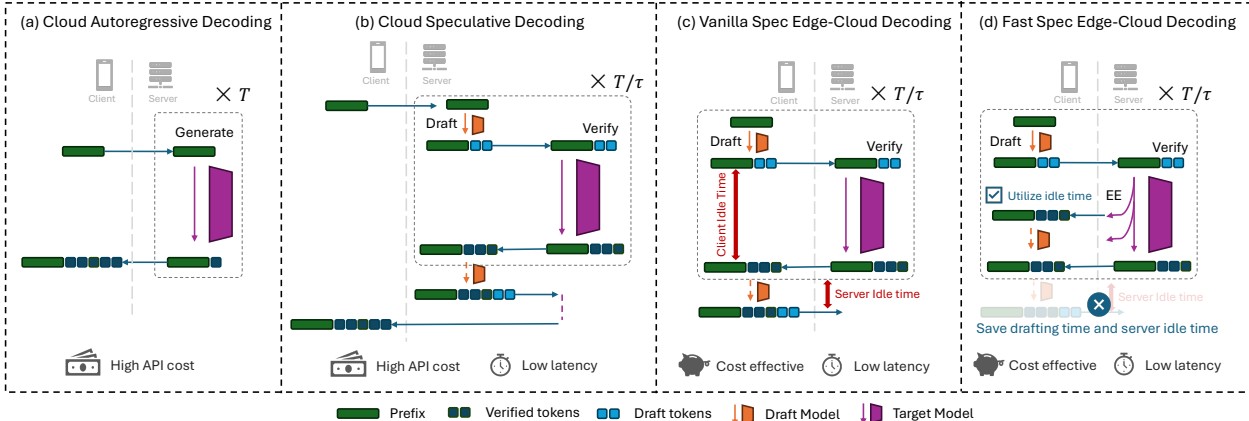

Figure 1: Illustration of traditional cloud-based autoregressive decoding versus cloud-based speculative decoding, vanilla speculative edge-cloud decoding, and the proposed preemptive drafting mechanism.

Table 1: Cost comparison between cloud autoregressive (AR) decoding and cloud speculative decoding (SD) and speculative edge-cloud decoding across different API providers on a set of candidate models. The measurement is based on 1 million requests, each consisting of 100 input tokens and 500 output tokens, assuming a draft length of $\gamma = 4$ tokens and an average of $\tau = 2.5$ accepted tokens per draft.

| API Provider | Draft/Target | Cost (In/Out per 1M tokens) | | API Cost | | |
|---|---|---|---|---|---|---|
| | | Draft | Target | Cloud AR | Cloud SD | Edge-cloud SD |
| Together AI | Qwen1.5-0.5B/Qwen1.5-72B | $0.1/$0.1 | $0.9/$0.9 | $540 | $360 (33% ↓) | $270 (**50%** ↓) |
| OpenRouter | llama-3.1-8b/llama-3.1-405b | $0.02/$0.05 | $0.9/$0.9 | $540 | $312 (42% ↓) | $270 (**50%** ↓) |
| Groq | llama3-8b-8192/llama3-70b-8192 | $0.05/$0.08 | $0.59/$0.79 | $454 | $286 (37% ↓) | $217 (**52%** ↓) |
| OpenRouter | Qwen-2-VL-7B/Qwen-2-VL-72B | $0.2/$0.2 | $0.7/$0.7 | $420 | $390 (7% ↓) | $210 (**50%** ↓) |

Gemma Team et al. (2024), driving breakthroughs in applications ranging from personalized assistants to complex problem-solving across various domains.

However, running large models is costly, requiring extensive computational resources for training, often spanning thousands of GPU hours, while inference at scale demands specialized hardware to maintain responsiveness Samsi et al. (2023). This creates significant barriers for smaller organizations and researchers who rely on expensive cloud-based APIs; for example, GPT-4.1 text generation costs $2.00/1M input tokens and $8.00/1M output tokens at the time of writing this paper.[1]

A potential solution is deploying LLMs on edge devices, which offers benefits like low latency, faster customization, and enhanced privacy in addition to cost-effectiveness. This is especially critical for real-time robotics applications, where decisions must be made on the fly, and the server cost can add up. For example, robotic platforms such as the Unitree Go2 quadruped are being equipped with language interfaces for real-world tasks like navigation, object interaction, and instruction following Cheng et al. (2024). However, such robots typically run on compute-constrained devices, making it infeasible to host large LLMs locally. For instance, the Unitree Go2 is powered by a Jetson Orin board with 16GB of unified memory, which is insufficient to run models over 10B parameters that require over 40GB of memory. Efficient decoding strategies like speculative decoding provide cost-effective solutions to bridge this gap. Speculative decoding Leviathan et al. (2023) uses a smaller model to generate tokens quickly, which are then verified by a larger model in parallel, significantly speeding up LLM inference. Despite its success on standalone machines, the application of speculative decoding on edge devices remains underexplored.

In this work, we propose a novel **speculative edge-cloud decoding method** to enable fast and cost-effective LLM inference at the edge. As shown in Fig. 1(a), traditional cloud-based autoregressive decoding takes a prompt from the client and performs $T$ forward passes on the target model to generate $T$ tokens, incurring

---

[1]OpenAI's API pricing as of May 2025.

an API cost proportional to $T$. Speculative decoding on the cloud (Fig. 1(b)) reduces target model calls by a factor of $\tau$, the number of tokens generated per draft-verify round. However, it introduces additional draft model calls which comes with a non-negligible cost Yan et al. (2024). Shifting drafting to the edge can eliminate this cost. Table 1 shows potential savings for example model pairs from various API providers.[2] Speculative edge-cloud decoding can reduce costs by up to 52% over cloud autoregressive decoding.

A straightforward edge speculation and cloud verification approach (Fig. 1(c)) suffers from inefficiencies: the client remains idle during server verification, and the server is unutilized while the client drafts tokens. To address this, we propose a *preemptive* drafting mechanism to maximize client-server utilization. As shown in Fig. 1(d), we introduce early exits in the target model to produce verified tokens before full verification. These early tokens enable the client to draft the next set preemptively, a process we call *pre-drafting*. If the final verification confirms the early tokens, the next set of draft tokens is readily available for verification, minimizing idle time and keeping both client and server continuously active. Our contributions are summarized as follows:

- We propose a novel framework that splits speculative decoding by hosting the draft model on the edge and the target model on the server, significantly reducing target model API costs.
- We introduce early exits in the target model to generate verified tokens ahead of full verification, enabling the client to preemptively draft the next tokens, minimizing idle time for both client and server.
- We conduct a comprehensive evaluation across 6 generation tasks on 3 sets of models. With Vicuna-68M as the draft model and Llama2-7B as the target model, we show an average 35% latency reduction from autoregressive to vanilla edge cloud speculative decoding and a further speedup of 11% with our fast decoding method.
- We demonstrate our approach on a real-world robotics platform (Unitree Go2 equipped with an NVIDIA Jetson Orin), highlighting the applicability of our method for enabling edge-cloud collaborative inference in embodied intelligence applications.

## 2 Background

**Speculative Decoding**: Speculative decoding follows a Draft-and-Verify approach, where each step starts with generating multiple candidate tokens, which are then verified by the target LLM in parallel, speeding up inference Leviathan et al. (2023). Formally, given an input prefix $x_{0:t}$, and a target model $\mathcal{M}_q$, a smaller draft model $\mathcal{M}_p$ generates the next $\gamma$ tokens $x_{t:t+\gamma}$ and their corresponding probability distribution $p_{t:t+\gamma}$ autoregressively:

$$x_{t:t+\gamma}, p_{t:t+\gamma} = \text{DRAFT}(\mathcal{M}_p, x_{0:t}) \tag{1}$$

The target model $\mathcal{M}_q$ verifies these tokens and decides how many to accept denoted by $\delta$ ($\delta \leq \gamma$), then produces the next token:

$$x_{t:t+\delta+1} = \text{VERIFY}(\mathcal{M}_q, x_{t:t+\gamma}, p_{t:t+\gamma}) \tag{2}$$

The process repeats with the input prefix extended to $t + \delta + 1$ and passed back to the draft model for the next round.

**Early Exit in Large Language Models**: Early exit strategies improve the efficiency of LLMs by terminating the generation process early if a sufficiently confident output is identified Panda et al. (2016); Chen et al. (2023). Given an LLM $\mathcal{M}$ with $L$ layers and an input sequence $x_{1:t}$, the hidden state at each layer $l$ is computed as:

$$h^{(l)} = f^{(l)}(h^{(l-1)}, x_{1:t}), \tag{3}$$

where $h^{(0)}$ is the input embedding. At each layer $l$, the model calculates logits by passing the hidden state through a language model (LM) head, denoted as $\mathbf{z}^{(l)} = \text{LMHEAD}(h^{(l)})$. It also computes a confidence score $S^{(l)}$ based on the softmax probability:

---

[2]API cost as of May 2025 based on https://www.helicone.ai/llm-cost.

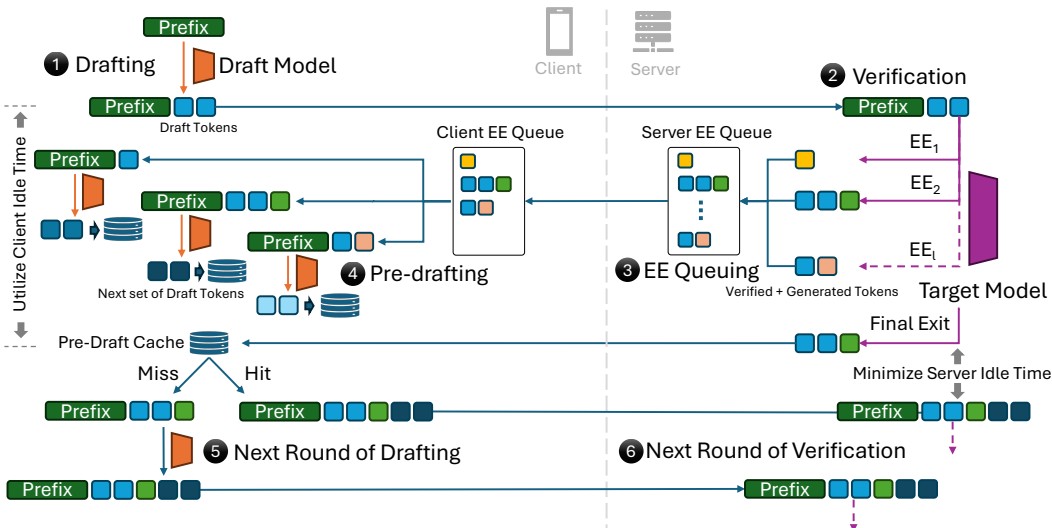

Figure 2: Illustration of our proposed approach. Given a prefix, the client generates two draft tokens and sends them to the server. The server verifies them using a target model with early exits, returning verified tokens and the next generated token. For each early exit, the client *pre-drafts* the next tokens and stores them in the pre-draft cache. If the final output matches a cache entry, the draft tokens are sent immediately, reducing latency.

$$S^{(l)} = \max\left(\text{softmax}(\mathbf{z}^{(l)})\right). \tag{4}$$

The model exits early at layer $l'$ if the confidence score exceeds a predefined threshold, $S^{(l')} \geq \tau$, and the next token $x_{t+1}$ is sampled from $\text{softmax}(\mathbf{z}^{(l')})$. We leverage this mechanism to generate early verified tokens in the target model, which are used to preemptively produce the next set of draft tokens.

## 3    Methodology

In our distributed speculative decoding setup, the client runs a lightweight *draft model*, denoted as $\mathcal{M}_p$, while the server hosts a large *target model* $\mathcal{M}_q$. As shown in Fig. 2, the algorithm takes the following sequence of steps:

**Step ①:** Given a prefix sequence $x_{0:t} = \{x_0, x_1, \ldots, x_t\}$, the client uses the draft model $\mathcal{M}_p$ to predict a sequence of $\gamma$ draft tokens (Eq. 1). These draft tokens $x_{t:t+\gamma}$, along with their probability distributions $p_{t:t+\gamma}$ are transmitted to the server for verification by the target model $\mathcal{M}_q$.

**Step ②:** The target model $\mathcal{M}_q$ is designed with multiple early exits, denoted as $\mathcal{M}_q^{(1:n)}$. Each early exit $i \in \{1, \ldots, n\}$ performs a verification step on the draft tokens (Eq. 2) and generates the next token. For example, if the early exit $i$ accepts $\delta^{(i)}$ tokens and generates the next token, the total generated tokens would be:

$$x_{t:t+\delta^{(i)}+1}^{(i)} = \text{VERIFY}(\mathcal{M}_q^{(i)}, x_{t:t+\gamma}, p_{t:t+\gamma}) \tag{5}$$

Here, $\delta^{(i)}$ denotes the number of draft tokens accepted by early exit $i$.

**Step ③:** Given that the communication channel is typically the bottleneck, early exit outputs are queued in the server's early exit queue as soon as they become available and are transmitted to the client sequentially.

**Step ④:** The client, in turn, stores the early exit outputs from the server in its own queue and processes each one in a new thread, *preemptively* generating the subsequent set of draft tokens for each early exit. This process is referred to as *pre-drafting*. For an early exit $i$, the newly verified/generated tokens from the server

Table 2: Early Exit training details. # Params and % Params denote the total number of trainable adapter parameters and their fraction compared to total model parameters respectively.

| Model | # Exits | # Params | % Params | Context | GPU Hours |
|---|---|---|---|---|---|
| lmsys/Vicuna-7B-v1.3 Zheng et al. (2023) | 31 | 101M | 1.48 | 1600 | 117 |
| lmsys/Vicuna-13B-v1.3 Zheng et al. (2023) | 39 | 158M | 1.20 | 800 | 122 |
| meta-llama/Llama-2-7B-hf Touvron et al. (2023) | 31 | 101M | 1.48 | 1600 | 119 |
| Qwen/Qwen2-VL-7B-Instruct Wang et al. (2024) | 27 | 88M | 1.02 | 1600 | 136 |

$x_{t:t+\delta^{(i)}+1}^{(i)}$ are concatenated with the original prefix $x_{1:t}$, resulting in a new prefix:

$$y_{0:t'}^{(i)} = \text{Concat}(x_{0:t}, x_{t:t+\delta^{(i)}+1}^{(i)}) \tag{6}$$

The *pre-draft* tokens represented as $y_{t':t'+\gamma}^{(i)}$ and their corresponding probabilities $p_{t':t'+\gamma}^{(i)}$, are then computed as:

$$y_{t':t'+\gamma}^{(i)}, p_{t':t'+\gamma}^{(i)} = \text{PreDraft}(\mathcal{M}_p, y_{0:t'}^{(i)}) \tag{7}$$

These pre-drafted tokens are subsequently stored in a cache referred to as the pre-draft cache.

**Steps ⑤ & ⑥:** Once the final output $x_{t:t+\delta+1}$ from the target model is received, the client checks whether these tokens were already processed in any of the early exits by looking at the pre-draft cache. If there is a hit, the corresponding pre-draft tokens are retrieved from the pre-draft cache and immediately sent to the server for the next round of verification, avoiding any delay. If it is a miss, a new set of draft tokens is generated following the usual drafting process. The server proceeds with the next round of verification over the new set of draft tokens.

This design enhances efficiency by leveraging the client's idle time for pre-drafting and reducing the server's idle time between verification rounds whenever there is a pre-draft cache hit. Importantly, the output is identical to that of standard speculative decoding since all tokens are verified at the final exit of the target model, guaranteeing no loss in accuracy. For detailed system design and pseudocode please refer to Appendix A.

**Early Exit Training**: We add adapter layers after each layer of the target model to train the early exits, as shown in Fig. 3. Each adapter connects to the language model (LM) head, and its loss is backpropagated to update only that adapter. This approach minimizes the number of trainable parameters and preserves the original model weights. Importantly, we do not define fixed confidence thresholds to determine when to exit; instead, all exit outputs are computed during inference, and their associated confidence scores are used to prioritize which outputs to communicate first.

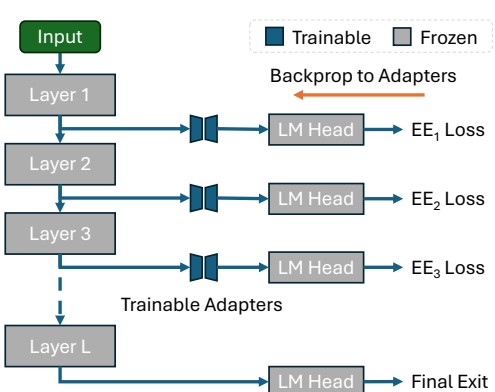

Figure 3: Illustration of training early exit adapters.

For language generation models, we train early exit adapters on the publicly available ShareGPT conversation dataset (hf:RyokoAI/ShareGPT52K) using a single NVIDIA A100 GPU with 80GB of VRAM. We fine-tune three models—Vicuna-7B, Vicuna-13B, and Llama2-7B—for 10 epochs each, using a batch size of 1 and a learning rate of 1e-4. Additionally, we train early exit adapters for a vision-language model based on Qwen2VL-7B using the Spacellava dataset (hf:remyxai/vqasynth_spacellava), which is generated using an open-source implementation of SpatialVLM Chen et al. (2024). Table 2 summarizes the number of early exits, total training time, and the number of trainable parameters for each model. Note that the context length was reduced during training to ensure compatibility with the memory limitations of a single A100 GPU.

Table 3: Notations used in our analysis.

| Notation | Description |
|---|---|
| $T_p$ : | Time for a single forward pass of the draft model $\mathcal{M}_p$ |
| $T_q$ : | Time for a forward pass of the target model $\mathcal{M}_q$ |
| $T_c$ : | Communication latency between the client and server |
| $c$ : | Latency ratio between the draft and target models ($T_p/T_q$) |
| $\gamma$ : | Number of draft tokens |
| $\tau$ : | Effective number of tokens generated per draft-verify round (# accepted tokens + 1 generated). |
| $n$ : | Total number of tokens generated |
| $r$ : | Cache miss rate |
| $T_r$ : | Latency of thread synchronization on cache hit |

## 4 Experiments

**Models and Benchmarks:** Following the standard speculative decoding literature Li et al. (2024), we evaluate our method on three model sets: Vicuna-68M/Vicuna-7B, Vicuna-160M/Vicuna-13B, and Vicuna-68M/Llama2-7B. We show the experiments on 6 standard generative task benchmarks spanning conversation Zheng et al. (2023), code generation Chen et al. (2021), mathematical reasoning Cobbe et al. (2021), instruction following Taori et al. (2023), summarization Nallapati et al. (2016), and question-answering tasks Kwiatkowski et al. (2019).

**Server Side Hardware**: We utilize a high performance computing cluster node equipped with a single A100 GPU with 80GB VRAM, 16 CPU cores, and 8GB of CPU memory per core as our server.

**Client Side Hardware:** We demonstrate our system on two types of client devices:

1. NVIDIA Jetson Nano: A compact AI development board tailored for edge computing. It includes a quad-core ARM Cortex-A57 CPU, a 128-core Maxwell GPU, and 4GB of LPDDR4 RAM shared between the CPU and GPU. With a performance of up to 472 GFLOPs, the Jetson Nano is ideal for edge applications.
2. Cluster Node with RTX 2080 Ti: This setup features a single RTX 2080 Ti GPU with 12GB VRAM, an 8-core CPU, and 4GB of RAM per core, providing a more powerful alternative for our experiments.

**Communication:** Communication between cluster nodes is facilitated by InfiniBand high-speed interconnect. Ethernet is used for communication between the cluster node and the Jetson Nano.

**Latency Calculation**: Table 3 summarizes the notations used in our analysis. Key variables include $T_q$, $T_p$, and $T_c$, representing latencies of the target model, draft model, and communication, respectively. Speculative decoding metrics include $\tau$, the effective tokens per draft-verification round. Hyperparameters are $n$, the total tokens, and $\gamma$, tokens per draft-verification round. Additional factors for our fast speculative decoding method include the cache miss rate $r$ and synchronization latency $T_r$ for cache hits. The latency calculation for autoregressive (AR) decoding, vanilla speculative edge-cloud decoding (SD), and our fast speculative edge-cloud decoding (FSD) are presented in Table 4. The latency of FSD method depends on the cache miss rate, $r$. In case of a cache hit, threads must synchronize, incurring a latency of $T_r$. Unless otherwise specified, we use $\gamma = 4$ and $n = 200$ in our experiments.

Table 4: Latency calculation. AR denotes cloud-based autoregressive decoding. SD and FSD refer to vanilla and fast speculative edge-cloud decoding, respectively.

| Method | Latency |
|---|---|
| Cloud AR | $2T_c + nT_q$ |
| Edge-Cloud SD | $\frac{n}{\tau}(2T_c + \gamma T_p + T_q)$ |
| Edge-Cloud FSD | $\frac{n}{\tau}(2T_c + r\gamma T_p + (1-r)T_r + T_q)$ |

## 4.1 System Metrics

We report the system metrics in Table 5, including drafting latency ($\gamma T_p$) at $\gamma = 4$, verification latency ($T_q$), the latency ratio $c$, and communication latency ($T_c$). On the Jetson Nano, drafting is about three times slower and communication twice as slow as on a cluster node with an RTX GPU. We report the maximum GPU VRAM and the number of early-exit threads supported by the system. On the RTX-equipped node, the system handles up to 30 threads for the Vicuna-68M model, but GPU VRAM (12 GB) limits the Vicuna-160M model to 15 threads before encountering an out-of-memory (OOM) error. On the Jetson Nano, both CPU threading and RAM are bottlenecks. The maximum number of threads is capped at 15 for the Vicuna-68M model, while the 4 GB memory limit allows only 7 threads for the Vicuna-160M model.

Table 5: Average system metrics that are dataset agnostic.

| Metric | Vicuna-68m/Vicuna-7B | | Vicuna-160m/Vicuna-13B | | Vicuna-68m/Llama2-7B | |
| --- | --- | --- | --- | --- | --- | --- |
| | Jetson | RTX | Jetson | RTX | Jetson | RTX |
| Drafting Latency ($\gamma T_p, \gamma = 4$) | 334ms | 102ms | 1596ms | 555ms | 301ms | 99ms |
| Verification Latency ($T_q$) | 497ms | 442ms | 616ms | 618ms | 522ms | 467ms |
| Latency Ratio ($c = T_p/T_q$) | 0.17 | 0.06 | 0.65 | 0.22 | 0.14 | 0.05 |
| Communication Latency ($T_c$) | 95ms | 42ms | 91ms | 46ms | 96ms | 47ms |
| Max GPU memory | 1.7G | 3.2G | 3.5G | 8.9G | 1.7G | 3.2G |
| Num EE Threads | 15 | 30 | 7 | 15 | 15 | 30 |

## 4.2 Speedup Results

**Evaluation Metrics:** Our fast decoding method with early exit is exact, with outputs identical to standard speculative decoding, ensuring **no loss in accuracy**. We define the following metrics to evaluate our method.

- **Speedup AR → SD**: Latency savings of vanilla speculative edge-cloud decoding (SD) compared to cloud based autoregressive (AR) baseline.
- **Speedup SD → FSD**: Latency savings of our fast speculative edge-cloud decoding (FSD) compared to the vanilla speculative edge-cloud decoding (SD).
- **Cache miss rate (lower the better)**: Frequency of cache misses, that indicates how often we fail to find the final output in one of the early exits.
- **Average Early Exit (lower the better)**: The average early exit that produces the same output as the final exit.

Table 6 presents the evaluation metrics on the benchmark datasets. In addition to the aforementioned evaluation metrics, the effective number of generated tokens per verification ($\tau$) is also reported. Note, $\tau$ remains identical to that of vanilla SD as our FSD method produces identical outputs but it highlights the reduction in API call costs.

**AR → SD:** On average, using RTX, vanilla SD achieves a 1.2x and 1.94x speedup over autoregressive decoding with the Vicuna-68M/Vicuna-7B and Vicuna-68M/Llama2-7B models, respectively. However, it results in a marginal slowdown with Vicuna-160M/Vicuna-13B. Jetson, being slower at drafting coupled with higher communication cost, reduces the speedup relative to autoregressive decoding, making it slower for Vicuna-68M/Vicuna-7B and Vicuna-160M/Vicuna-13B models, though it achieves a 1.34x speedup with Vicuna-68M/Llama2-7B. The speedup from autoregressive to SD primarily depends on $c$, $\tau$, and $T_c$, with ideally requiring low values of $c$ and $T_c$ and a high $\tau$. Since Jetson has a high $c$ and $T_c$, it underperforms compared to autoregressive on Vicuna-68M/Vicuna-7B and Vicuna-160M/Vicuna-13B models. In contrast, for Vicuna-68M/Llama2-7B, a lower $c$ combined with a higher $\tau$ yields a 1.35x speedup.

**SD → FSD:** Our FSD provides consistent speedup over vanilla SD across all datasets on both RTX and Jetson. On the RTX client, it achieves average speedups of 1.06x, 1.10x, and 1.06x for Vicuna-68M/Vicuna-7B, Vicuna-160M/Vicuna-13B, and Vicuna-68M/Llama2-7B, respectively. Similarly, on Jetson Nano, the speedups are 1.04x, 1.06x, and 1.11x for the same model pairs. The primary benefit of FSD lies in its *pre-drafting*

Table 6: Speedup evaluation on standard language benchmark datasets.

| Benchmark | Metric | Vicuna-68m/Vicuna-7B | | Vicuna-160m/Vicuna-13B | | Vicuna-68m/Llama2-7B | |
|---|---|---|---|---|---|---|---|
| | | Jetson | RTX | Jetson | RTX | Jetson | RTX |
| MT-bench | Speedup AR $\rightarrow$ SD | 0.70x | 1.30x | 0.42x | 0.97x | 1.34x | 2.01x |
| | Speedup SD $\rightarrow$ FSD | 1.04x | 1.04x | 1.05x | 1.09x | 1.07x | 1.02x |
| | Avg Tokens $\tau$ | 2.30 | 2.01 | 2.28 | 2.98 | 4.12 | 3.48 |
| | Cache miss rate | 60.92% | 20.07% | 62.49% | 16.40% | 36.73% | 13.38% |
| | Avg EE | 8 | 13 | 9 | 15 | 9 | 11 |
| HumanEval | Speedup AR $\rightarrow$ SD | 0.79x | 1.42x | 0.47x | 0.85x | 1.53x | 2.09x |
| | Speedup SD $\rightarrow$ FSD | 1.04x | 1.03x | 1.06x | 1.15x | 1.22x | 1.07x |
| | Avg Tokens $\tau$ | 2.04 | 2.66 | 2.14 | 2.04 | 4.16 | 3.83 |
| | Cache miss rate | 64.73% | 15.79% | 57.64% | 25.16% | 17.6% | 1.75% |
| | Avg EE | 7 | 16 | 8 | 14 | 4 | 3 |
| GSM8K | Speedup AR $\rightarrow$ SD | 0.63x | 1.11x | 0.40x | 0.77x | 1.15x | 1.69x |
| | Speedup SD $\rightarrow$ FSD | 1.04x | 1.08x | 1.06x | 1.13x | 1.07x | 1.06x |
| | Avg Tokens $\tau$ | 2.08 | 1.96 | 2.23 | 2.29 | 3.48 | 3.29 |
| | Cache miss rate | 61.21% | 12.40% | 58.04% | 18.37% | 41.87% | 9.55% |
| | Avg EE | 8 | 13 | 9 | 14 | 7 | 10 |
| Alpaca | Speedup AR $\rightarrow$ SD | 0.63x | 1.06x | 0.42x | 0.74x | 1.42x | 1.99x |
| | Speedup SD $\rightarrow$ FSD | 1.04x | 1.07x | 1.05x | 1.12x | 1.10x | 1.18x |
| | Avg Tokens $\tau$ | 2.09 | 1.96 | 2.32 | 2.36 | 4.29 | 3.62 |
| | Cache miss rate | 64.60% | 19.29% | 60.94% | 25.45% | 36.60% | 4.28% |
| | Avg EE | 8 | 14 | 8 | 14 | 9 | 2 |
| CNN/DM | Speedup AR $\rightarrow$ SD | 0.72x | 1.20x | 0.38x | 0.73x | 1.41x | 1.91x |
| | Speedup SD $\rightarrow$ FSD | 1.03x | 1.07x | 1.03x | 1.07x | 1.07x | 1.04x |
| | Avg Tokens $\tau$ | 2.32 | 1.95 | 2.08 | 2.10 | 4.32 | 3.43 |
| | Cache miss rate | 70.70% | 29.40% | 69.92% | 38.09% | 47.13% | 13.97% |
| | Avg EE | 10 | 15 | 8 | 16 | 14 | 13 |
| NQ | Speedup AR $\rightarrow$ SD | 0.65x | 1.10x | 0.46x | 0.82x | 1.26x | 1.93x |
| | Speedup SD $\rightarrow$ FSD | 1.02x | 1.06x | 1.04x | 1.12% | 1.05x | 1.01x |
| | Avg Tokens $\tau$ | 2.08 | 2.05 | 2.44 | 2.62 | 3.83 | 3.62 |
| | Cache miss rate | 71.50% | 22.59% | 63.25% | 30.25% | 57.11% | 14.88% |
| | Avg EE | 8 | 14 | 8 | 15 | 13 | 14 |
| **Average** | **Speedup AR $\rightarrow$ SD (↑)** | 0.69x | 1.20x | 0.42x | 0.94x | 1.35x | **1.94x** |
| | **Speedup SD $\rightarrow$ FSD (↑)** | 1.04x | 1.06x | 1.05x | 1.10x | **1.11x** | 1.06x |
| | **Avg Tokens $\tau$ (↑)** | 2.15 | 2.10 | 2.25 | 2.40 | **4.03** | 3.54 |
| | **Cache miss rate (↓)** | 65.61% | 19.59% | 62.05% | 27.95% | 39.94% | **9.63%** |
| | **Avg EE (↓)** | **8** | 14 | **8** | 14 | 9 | 9 |

mechanism, which enables these improvements over vanilla SD. This mechanism's impact is reflected in the cache miss rate, indicating how often the final output is available through early exits, allowing pre-drafting of the next set of tokens. The average early exit metric further highlights how quickly verified tokens are obtained, enabling efficient generation of subsequent draft tokens.

## 4.3 Ablation Studies

**Effect of Number of Threads:** In Figure 4a, we show the speedup of our FSD relative to vanilla SD and the cache miss rate as the number of early exit threads increases up to 30. Using the GSM8K dataset with the Vicuna-68M/Vicuna-7B models on an RTX client, we find that the cache miss rate decreases as more threads are added, improving speedup. However, after around 15 threads, the speedup begins to plateau, and further increases in thread count yield minimal additional speedup. This is because the priority queues process the most promising early exits first, making it more likely to match the final output with the initial threads rather than the later ones.

**Effect of $\gamma$:** The number of tokens, $\gamma$, significantly influences the efficiency of speculative decoding. In Fig. 4b, we plot speedup between SD and FSD, and between autoregressive (AR) and SD, as $\gamma$ increases up to 10. We use the GSM8K dataset with the Vicuna-68M/Vicuna-7B models on an RTX-based client. Our FSD method shows greater latency improvements compared to vanilla SD as $\gamma$ increases, enhancing the benefits of

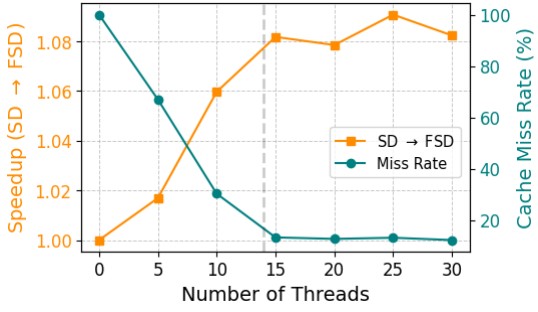

(a) Effect of number of early exit threads.   (b) Effect of number of draft tokens ($\gamma$).

Figure 4: Ablation studies: (a) Effect of varying the number of early exit threads, (b) Effect of varying the number of draft tokens ($\gamma$).

pre-drafting. However, an excessively large $\gamma$ can hinder the speculative decoding process, causing the overall speedup to decrease. As shown, the speedup of SD relative to AR falls below 1x when $\gamma$ exceeds 5.

**Importance of Priority Queue:** Since our system is asynchronous, we need queues for graceful operation. Further, we organize the queues in priority, determined by the confidence score of the generated token (Eq. 4). This prioritization is especially beneficial when the number of threads is limited. Table 7 presents an ablation study comparing different queue configurations. On the server side, we use either a priority queue or a FIFO queue, while on the client side, we also include a random queue as an option. We report the cache miss rate ($r$) for systems with 3, 5, and 10 threads, denoted as 3T, 5T, and 10T, respectively.

Table 7: Ablation study of client-server queue strategies. $r$ indicates the average response time (lower is better) for different queue lengths.

| Client | Server | $r$ (3T) | $r$ (5T) | $r$ (10T) |
|---|---|---|---|---|
| Priority | Priority | **79.85** | **62.93** | 27.57 |
| | FIFO | 80.40 | 63.48 | 27.73 |
| Random | Priority | 82.32 | 63.25 | **26.92** |
| | FIFO | 84.56 | 65.36 | 27.68 |
| FIFO | Priority | 82.61 | 64.43 | 27.69 |
| | FIFO | 85.43 | 65.67 | 27.84 |

Our findings indicate that when the server uses a priority queue, it significantly improves performance for any given queue type on the client, although this benefit decreases with a higher thread count. On the client side, a priority queue consistently outperforms both the random and FIFO queues.

## 4.4 Robotics Case Study: Vision-Language Navigation on Unitree Go2

To evaluate the real-world applicability of our approach, we deploy the edge-cloud speculative decoding system on the Unitree Go2 EDU quadruped robot. This platform features an onboard NVIDIA Jetson Orin board, which includes an 8-core ARM Cortex-A78AE v8.2 64-bit CPU and 16GB of 128-bit LPDDR5 unified memory, offering up to 157 TOPS of compute. Communication between the robot and the server is established over Wi-Fi 6.

The robot receives natural language instructions such as "go to the red chair" or "turn left at the hallway" and uses its front-facing RGB camera to perceive the environment. A vision-language model (VLM) processes the visual observations and language commands to generate mid-level navigation actions (e.g., move forward small/medium/large), following the approach of Cheng et al. (2024). These actions are then executed by the robot's onboard controller. To enhance decision quality and interpretability, we additionally prompt the VLM to provide reasoning alongside its action outputs.

We deploy a quantized version of Qwen-2-VL-2B as the on-device draft model and offload token verification to the full-size Qwen-2-VL-7B model hosted on an A100 GPU in the cloud. Figure 5 illustrates an example scenario in which the robot is instructed to locate a specific object—in this case, a silver bottle. To test the

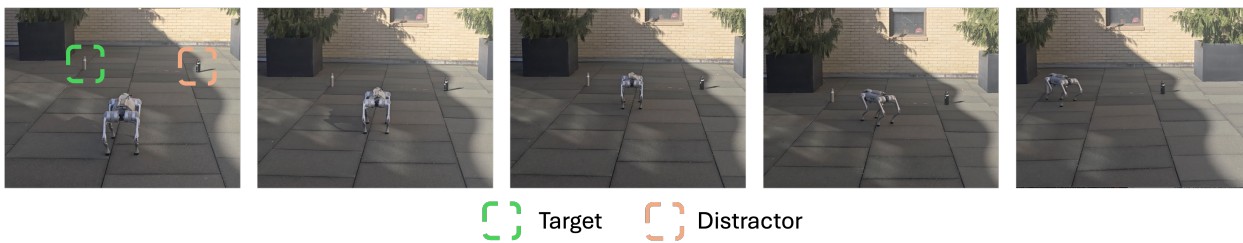

Figure 5: Example run of the Unitree Go2 robot performing an object-finding task using vision-language-based control. The robot receives the instruction *"find the silver bottle"* and navigates the environment while distinguishing the correct object from a similar distractor.

Table 8: System-level evaluation metrics for our edge-cloud speculative decoding setup on the Unitree Go2 robot (Jetson Orin) and A100 server. (a) Reports core system and latency metrics. (b) Summarizes the performance gains with our FSD method.

| Metric | Value |
|---|---|
| Drafting Latency ($\gamma T_p$, $\gamma = 4$) | 288ms |
| Verification Latency ($T_q$) | 620ms |
| Latency Ratio ($c = T_p/T_q$) | 0.11 |
| Communication Latency ($T_c$) | 120ms |
| Max GPU Memory | 12.4G |
| Num EE Threads | 6 |

(a) System and latency metrics

| Metric | Value |
|---|---|
| Avg Tokens ($\tau$) ($\uparrow$) | 2.92 |
| Cache Miss Rate ($\downarrow$) | 55.63 |
| Avg EE ($\downarrow$) | 8 |
| Speedup AR $\rightarrow$ SD ($\uparrow$) | 0.90x |
| Speedup SD $\rightarrow$ FSD ($\uparrow$) | **1.34x** |

(b) FSD speedup and performance metrics

model's reasoning and grounding capabilities, we introduce a distractor object of similar appearance. The robot successfully navigates the environment and identifies the correct object, demonstrating the effectiveness of our method on a vision-language-based control task.

Table 8(a) reports key system-level metrics from our deployment, including drafting and verification latencies, communication overhead, and peak GPU memory usage. Table 8(b) highlights the performance improvements enabled by our method, showing speedups from standard autoregressive decoding (AR) to edge-cloud speculative decoding (SD) and further to Fast Speculative Decoding (FSD). It also includes average accepted tokens per round, cache miss rate, and average early exits. Overall, our system achieves a **21%** speedup over conventional cloud-based autoregressive decoding, validating the practicality of our approach for real-time, language-conditioned robot control on resource-constrained edge platforms.

## 5 Related Work

There is a significant interest in enabling edge devices to run LLMs. The deployment of collaborative AI inference systems across the edge and the cloud introduces unique challenges such as latency constraints, bandwidth limitations, and inconsistent network conditions. One straightforward approach is to design smaller models Lu et al. (2024). While all the popular class of models such as OPT Zhang et al. (2022), Llama Touvron et al. (2023); Dubey et al. (2024), and Gemma Team et al. (2024) have smaller scale models, they are either not small enough to run on an edge device or not accurate enough to reliably deploy in practical applications. Quantization is one of the heavily focused methods to enable on-device LLMs Lin et al. (2024). Yu et al. (2024) aim to compress the models with layer-wise pruning and quantization to enable edge LLMs. Qu et al. (2024) discuss an approach of enabling LLMs to run on 6g edge devices. On system side, Xu et al. (2024a) focus on leveraging on-device Neural Processing Unit (NPU).

Early exit strategies, which allow intermediate layers of deep networks to make predictions without waiting for the full forward pass, have been extensively explored for resource-constrained devices. Pioneering works such

as Conditional Deep Learning Panda et al. (2016) and BranchyNet Teerapittayanon et al. (2016) introduced the idea of adding multiple exit points to deep neural networks to reduce computation time. Recent research has also explored layer skipping in LLMs for enhanced efficiency Fan et al. (2024), with dynamic compute allocation based on tokens Raposo et al. (2024). In terms of multi-device speculative decoding, McDanel (2024) has recently shown that asynchronous speculative decoding over multiple GPUs can be beneficial. However, it uses shared memory to communicate between devices, so it is not directly applicable to edge-cloud scenarios. To the best of our knowledge, this is the first work to show end-to-end speculative decoding with models split between edge and cloud. Further, we comprehensively analyze and demonstrate the system-level trade-offs during the implementation of collaborative edge-cloud decoding, which no prior work has investigated.

## 6 Conclusion and Discussion

We introduced a novel speculative edge-cloud decoding framework, offering a cost-effective alternative to traditional cloud-based deployment. By distributing the draft and target models between edge and server environments, our solution significantly reduces high API costs. Early exits and pre-drafting allow us to enhance parallelism by leveraging idle client time and reducing server idle time. Our comprehensive end-to-end evaluation on the NVIDIA Jetson Nano highlights the feasibility of efficient edge-cloud collaborative LLM inference on resource-limited edge devices. On Jetson Nano, speculative edge-cloud decoding achieves up to a 35% speedup over cloud-based autoregressive decoding, with up to an additional 11% performance gain enabled by pre-drafting and early exits. Further, we validate our approach with execution of vision language models on the Unitree Go2 quadruped robot. We achieve an overall 21% speedup over standard cloud-based autoregressive decoding, demonstrating the effectiveness and real-world applicability of our framework for robotics use cases.

Our method operates effectively without making assumptions about communication delays, and we show that it remains practical under real-world conditions. However, in extreme scenarios—such as round-trip latencies exceeding 300–400 ms or highly variable throughput on congested mobile networks—our performance gains may diminish. Our use of priority queues is specifically designed to address these challenges by optimizing bandwidth utilization and maintaining responsiveness under constrained network conditions. Future work could explore adaptive communication strategies and dynamic scheduling policies to enhance robustness. Further, we believe that continued advancements in edge-cloud infrastructure—including lower-latency connectivity (e.g., 5G/6G), transport protocol optimizations, and hardware-accelerated interconnects—will reduce this bottleneck.

Additionally, while pre-drafting leverages idle client time for parallelism, it introduces a compute overhead on the client. This is manageable for most edge platforms, and the ability to scale the number of threads provides a flexible trade-off between latency and energy efficiency, especially for battery-powered devices. That said, we acknowledge that our approach introduces additional system complexity. Coordinating early exits, managing pre-draft caches, and maintaining token-level priority queues require careful orchestration. Although the performance gains justify this complexity in many settings, future work could explore lightweight abstractions or runtime systems to streamline deployment. Another practical consideration is the training overhead introduced by early exit adapters. Fine-tuning these exit heads on target models incurs a one-time cost (approximately 100 GPU hours per model). Although the training cost is a one-time expense that can be amortized over repeated use, it can limit the frequency with which the model can be updated in practice. Our results are also sensitive to the quality of the early exit calibration. High exit misprediction or cache miss rates can degrade throughput and reduce pipelining efficiency. Adaptive confidence thresholds or dynamic calibration methods may mitigate this issue and offer further robustness.

Cache miss rates significantly affect the efficiency of speculative decoding and are primarily influenced by two factors: (i) the quality of the early exit adapters and (ii) the computational capacity of the client device. Well-trained and calibrated early exits are more likely to produce outputs that match the final model, thereby increasing cache hit rates and improving decoding throughput. However, on resource-constrained devices like the Jetson Nano, limited memory and compute capabilities restrict the number of early exit outputs that can be computed and stored in the pre-draft cache. Although we train early exit adapters on general-purpose conversational data, task-specific fine-tuning can improve their alignment with final outputs, thereby reducing

cache misses. On the client side, we prioritize speculative drafts from higher-confidence exits. However, these typically correspond to later layers, which are delayed in arriving to the client. Future extensions could explore more sophisticated scheduling strategies beyond static confidence-based prioritization, potentially introducing controlled delays to improve hit rates while balancing latency and resource usage.

Our proof-of-concept is implemented in Python, which offers ease of experimentation but leaves room for performance optimization. A low-level C++ implementation with multithreading and shared memory could significantly enhance efficiency, particularly for latency-sensitive applications. While the current system supports only single-client interaction with the server, extending it to handle multi-client concurrency is a natural next step. This introduces new challenges, such as synchronizing early exit queues, managing server resource contention, and ensuring fair allocation of communication bandwidth across clients. Future iterations may incorporate techniques like batched verification and client prioritization policies to improve scalability and support robust, concurrent edge-cloud inference with early exits. Future work could explore a broader range of edge-cloud configurations beyond the Jetson Nano and A100 GPU used in this evaluation. This includes ultra-low-power microcontrollers, ARM-based edge devices, and emerging robotic platforms. On the cloud side, potential directions include evaluating newer accelerators such as Google TPUs, Groq LPUs, and Cerebras WSEs. It would also be valuable to assess performance under diverse network conditions, including 4G and 5G connectivity.

**Broader Impact**: FSD reduces reliance on always-on cloud infrastructure by offloading a portion of the inference workload to the edge, thereby lowering energy consumption per query. As a reference point, the NVIDIA A100 GPU has a thermal design power (TDP) of 250–400W, whereas the Jetson Nano consumes only 5–10W—nearly two orders of magnitude less. This makes the additional power required by the edge device negligible relative to the server. Consequently, even partial offloading can lead to significant energy savings; for instance, our approach reduces server utilization by approximately 50%, potentially halving the associated power consumption. While we do not directly measure carbon emissions, this reduction in centralized compute contributes to a smaller environmental footprint. Although our evaluation is based on NVIDIA hardware, the FSD framework is both model- and hardware-agnostic, and can be adapted to open-source models and alternative edge platforms that support multi-threading or multi-processing. We do not anticipate any hardware-specific dependencies or exclusionary practices resulting from our framework.

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

## A  System Design

**Client:** On the client side, the primary goal is to maximize idle time usage and increase cache hit rates. As shown in Algorithm 1, the client maintains a priority queue $Q_p$, a pre-draft cache $C$, and a RECEIVER thread. After sending draft tokens to the server for verification, the RECEIVER thread listens for server callbacks, which provide outputs from early exits.

Since the client's bottleneck lies in the processing power required for generating pre-draft tokens, it is essential to prioritize the handling of early exit outputs. The priority queue $Q_p$ organizes these outputs according to their confidence levels (Eq. 4), prioritizing the most promising ones for pre-drafting. It is populated asynchronously as early exit outputs are received by the client. If the client's device has multiple available threads, it can process several early exits in parallel to generate more pre-draft tokens. All the pre-draft tokens are stored in the pre-draft cache. Once the client receives the final exit output, it checks the pre-draft cache for the corresponding tokens. If there is a cache hit, the pre-drafted tokens are sent to the server immediately for the next verification round.

**Server:** As detailed in Algorithm 2. the server consists of two asynchronous threads: LISTENER and SENDER. The LISTENER processes the verification requests from the client. It takes in the prefix $x_{1:t}$, draft tokens $x_{t:t+\gamma}$, and their corresponding probability distribution $p_{1:\gamma}$.

As shown in Fig. 2, communication typically becomes the bottleneck in the server as early exit outputs are produced faster than the network can transmit. Early exit outputs are placed in a queue on the server side and transmitted sequentially to handle this. Let $Q_e$ represent the server-side queue storing the early exit outputs:

$$Q_e = \{x_{t:t+\delta^{(1)}+1}^{(1)}, \ldots, x_{t:t+\delta^{(L)}+1}^{(L)}\}. \tag{8}$$

Asynchronously, the SENDER thread sends the early exit outputs from the queue based on priority determined by the confidence score (Eq. 4).

## B  Speedup Projection Analysis

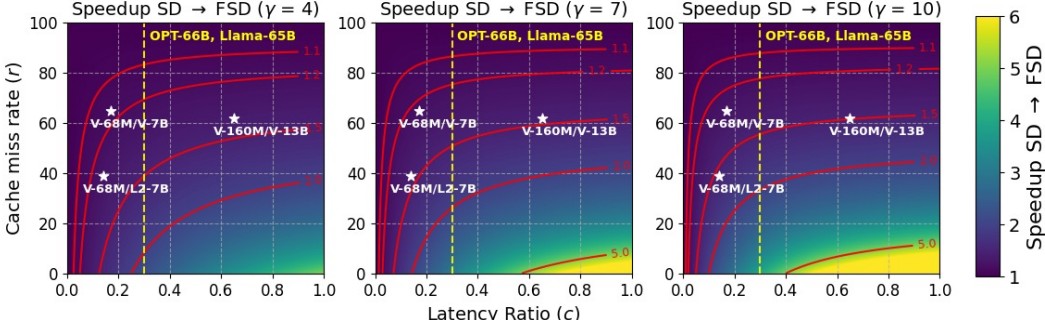

Figure 6: Estimated speedup per round of our FSD relative to vanilla SD method assuming no communication latency. We emphasize the contours for speedups of 1.1x, 1.2x, 1.5x, 2x, and 5x, and we indicate the position of our models within this landscape. Additionally, we highlight the operating range for the larger models OPT-66B and Llama-65B.

One of the bottlenecks in our system is communication between the devices. In addition to directly adding to the latency, it also bottlenecks the number of early exit verifications communicated back to the client.

---

**Algorithm 1** Client-Side Algorithm

---

1: **Initialize:** Draft model $\mathcal{M}_p$, Queue $Q_p$, Cache $C$, Receiver
2: **Input:** Prefix $x_{1:t}$, # Total Tokens $T$, # Draft tokens $\gamma$
3: **Output:** Final tokens $x_{t+1:T}$
4: **for** $i = 1$ to $\gamma$ **do**
5:      $x_{t+i}, p_i \leftarrow \text{Draft}(\mathcal{M}_p, x_{1:t+i-1})$
6: **end for**
7: $\text{Send}(x_{1:t+\gamma}, p_{1:\gamma})$
8: **while** $t < T$ **do**
9:      **while** $Q_p$ not empty **do**
10:          $x'_{t+1:t+\delta'+1}, s' \leftarrow Q_p.\text{pop}()$
11:          **if** $x'_{t+1:t+\delta'+1}$ not in $C$ **then**
12:              $C[x'_{t+1:t+\delta'+1}] \leftarrow \text{PreDraft}(x_{1:t}, x'_{t+1:t+\delta'+1})$
13:          **end if**
14:      **end while**
15:      $y_{1:t'} \leftarrow \text{Concat}(x_{1:t}, x_{t+1:t+\delta+1})$
16:      **if** $x_{t+1:t+\delta+1} \in C$ **then**
17:          $y_{t':t'+\gamma}, p_{1:\gamma} \leftarrow C[x_{t+1:t+\delta+1}]$
18:      **else**
19:          **for** $i = 1$ to $\gamma$ **do**
20:              $y_{t'+i}, p_i \leftarrow \text{Draft}(\mathcal{M}_p, y_{1:t'+i-1})$
21:          **end for**
22:      **end if**
23:      $\text{Send}(y_{t':t'+\gamma}, p_{1:\gamma})$
24:      $t \leftarrow t', x \leftarrow y, C.\text{reset}(), Q_p.\text{reset}()$
25: **end while**

26: **function** PreDraft
27:      **Input:** Prefix $x_{1:t}$, Tokens $x'_{t+1:t+\delta'+1}$
28:      $y_{1:t'} \leftarrow \text{Concat}(x_{1:t}, x'_{t+1:t+\delta'+1})$
29:      **for** $i = 1$ to $\gamma$ **do**
30:          $y_{t'+i}, p'_i \leftarrow \text{Draft}(\mathcal{M}_p, x'_{1:t'+i-1})$
31:      **end for**
32:      **return** $y_{t':t'+\gamma}, p'_{1:\gamma}$
33: **end function**

34: **function** Receiver
35:      **Input:** Tokens $x'_{t+1:t+\delta'+1}$, Priority $s'$, *isfinalexit*
36:      **if** *isfinalexit* **then**
37:          $x_{t+1:t+\delta+1} \leftarrow x'_{t+1:t+\delta'+1}$
38:      **else**
39:          $Q_p.\text{push}(x'_{t+1:t+\delta'+1}, s')$
40:      **end if**
41: **end function**

---

This increases the cache miss rate further increasing the latency. While this is a challenge at present due to limited communication network capabilities, several works have shown the vision of having edge LLMs on 6g networks with a projected network speed up to 10 Tbps Banafaa et al. (2023); Lin et al. (2023); Xu et al. (2024b); Friha et al. (2024); Qu et al. (2024); Zhang et al. (2024). In scenarios where communication latencies $T_c$ is negligible relative to drafting and verification time, and ignoring thread synchronization latency ($T_r$) for simplicity, we can approximate the speedup SD → FSD as:

$$\text{Speedup (SD} \rightarrow \text{FSD)} = \frac{\gamma c + 1}{r(\gamma c) + 1} \tag{9}$$

---

**Algorithm 2** Server-Side Algorithm

---

1: **Initialize:** Target Model $\mathcal{M}_q$, Verification criterion Verify, Queue $Q_e$, Listener, Sender
2: **function** Listener
3:     **Input:** Prefix and draft tokens $x_{1:t+\gamma}$, Probs. $p_{1:\gamma}$
4:     $x_{t+1:t+\delta+1}^{(1:L)}, q_{1:\delta+1}^{(1:L)} \leftarrow \text{Verify}(\mathcal{M}_q, x_{1:t+\gamma}, p_{1:\gamma})$
5:     **for all** $i = 1, \ldots, L-1$ **do**
6:         $s^{(i)} \leftarrow \max(q_{1:\delta+1}^{(i)})$
7:         $Q_e.\text{push}(x_{t+1:t+\delta+1}^{(i)}, s^{(i)})$
8:     **end for**
9:     $\text{Send}(x_{t+1:t+\delta+1}^{(L)}, \text{isfinalexit} = \text{True})$
10:    $Q_e.\text{reset}()$
11: **end function**
12: **function** Sender
13:     **while** $Q_e$ not empty **do**
14:         $(x'_{t+1:t+\delta+1}, s') \leftarrow Q_e.\text{pop}()$
15:         $\text{Send}(x'_{t+1:t+\delta+1}, s', \text{isfinalexit} = \text{False})$
16:     **end while**
17: **end function**

---

This formula reduces the final speedup to be affected by two factors—Cache miss rate $r$ and Latency Ratio $c$. Cache miss rate $r$ depends on the redundancy in the target model and how well the early exit adapters are trained. On the other hand, $c$ is highly dependent on the compute capability of the edge device. Since edge devices are often slower, this pushes the $c$ to be higher.

We visualize Eq. 9 as a heatmap in Fig. 6 for $\gamma$ values 4, 7 and 10. For reference, we plot the measured $c$ and $r$ values based on the Jetson implementation of our current set of models within this landscape. Naturally, having a lower $r$ will improve speedup, but the usefulness of our FSD method becomes more pronounced as we get to higher $c$ and $\gamma$. For model sets with a latency ratio greater than 0.5 and well-trained early exit adapters that achieve a cache miss rate of less than 10%, we can anticipate a speedup over 5x.

Extending our analysis to larger models, specifically OPT-66B and Llama-65B with draft models OPT-125M and NoFT-Wide-796M, we use reported latencies from Yan et al. (2024) (6.6 ms for draft, 67 ms for target) and factor in a 3x slowdown on Jetson, arriving at $c \approx 0.3$. This value is illustrated by the yellow line in Fig. 6. For instance, to achieve a 2x speedup with $\gamma$ values of 4, 7, and 10, the cache miss rate must remain below 10%, 25%, and 35%, respectively.

## C  Batch Processing

Table 9 shows latency analysis for Vicuna-7B (A100) and Vicuna-68M (Jetson Nano). Batch processing improves throughput but not always latency; e.g., batch size 32 increases A100 latency by over 5x. However, API providers often offer discounts for batch processing (e.g., OpenAI provides 50% discount OpenAI Pricing), making it a cost-saving approach. On the client, batch processing shows a smaller latency increase—batch sizes of 1 and 8 differ by 15%. A batched pre-drafting approach could reduce latency but requires waiting to accumulate multiple early exits, introducing a trade-off.

## D  Early Exit Analysis

In Fig. 7, we present a detailed comparison of token match percentages and average confidence scores across early exit layers in the Vicuna-7B model. The token match percentage quantifies the proportion of tokens generated by each early exit that match the corresponding tokens in the final model output. As expected, this match percentage increases steadily with deeper exits, indicating that later layers are more aligned with the final output distribution. Interestingly, the match percentage grows sharply beyond the halfway point of the network, suggesting that substantial refinement of predictions occurs in the latter half of the model's

Table 9: Server and client latency for different batch sizes for Vicuna-68M and Vicuna-7B models on client and server respectively.

| Batch Size | Server Latency (A100) | Client Latency (Jetson) |
|---|---|---|
| 1 | 17.94 | 7.76 |
| 2 | 21.75 | 8.02 |
| 3 | 24.35 | 7.57 |
| 4 | 24.63 | 7.87 |
| 8 | 30.73 | 8.97 |
| 12 | 44.52 | 9.71 |
| 16 | 45.79 | 10.63 |
| 32 | 96.94 | 14.49 |
| 64 | 258.79 | OOM |

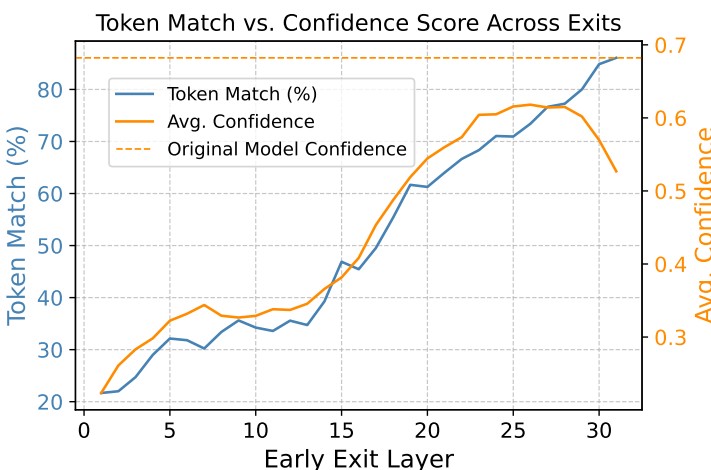

Figure 7: Token match percentage and Confidence Score across early exits measured with Early Exit adapters trained on the Vicuna-7B model.

depth. Notably, early exits in the initial layers achieve only 20–30% token-level agreement with the final model, underscoring the limited utility of very shallow exits for high-fidelity generation.

Alongside match rates, we also plot the average confidence score at each early exit, defined as the softmax probability of the most likely token at each timestep. While confidence generally correlates with layer depth, we observe that confidence saturates earlier than match accuracy. This suggests that some early exits, particularly in the mid-range layers, can achieve high token match percentages despite lower confidence scores—indicating that accurate predictions may be made even before the model's certainty peaks. Such behavior can be leveraged for adaptive inference: exits that combine high match rates with sufficient confidence offer promising points for early stopping, enabling latency reduction without substantial loss in output quality. These findings highlight the value of jointly analyzing confidence and output fidelity when designing efficient decoding strategies based on early exits.

