# OpenReview forum: "Fast and Cost-effective Speculative Edge-Cloud Decoding with Early Exits"
_TMLR — Accepted by TMLR_

### Review · Reviewer_ni9w · 2025-06-13

**Summary Of Contributions:**

This paper introduces a novel framework for fast and cost-effective LLM inference. The core idea is to run a small draft model on the edge device to generate candidate tokens, while a large target model on the server verifies these tokens via early exit. A key contribution is the introduction of early exits in the target model, allowing partially verified tokens to be sent back to the client sooner. This enables the client to pre-draft subsequent tokens preemptively, utilizing its idle time and enhancing parallelism. The authors demonstrate significant latency reductions (up to 35% + 11%) compared to traditional cloud-based autoregressive decoding and vanilla edge-cloud speculative decoding.

**Audience:**

Yes

**Claims And Evidence:**

Yes

**Requested Changes:**

1. Discuss System Complexity. Briefly acknowledge the increased system complexity (e.g., managing pre-draft caches, priority queues) in the Discussion section. Also, comment on the sensitivity of FSD's gains to the early exit quality and cache miss rate, perhaps suggesting scenarios or future work to improve early exit calibration or mitigate high miss rates.

2. Address Training Overhead and Communication: In the Discussion or Future Work, briefly mention the early exit training overhead as a practical consideration. Also, touch upon how further advancements in edge-cloud communication infrastructure could unlock even greater potential for the proposed FSD method.

**Strengths And Weaknesses:**

Strengths:

1. Cost Reduction. The proposed method intelligently utilizes client idle time and server resources, leading to significant demonstrated reductions in both inference latency and operational API costs.

2. The method's effectiveness is robustly evaluated across multiple LLM pairs, standard benchmarks, and different client hardware, all while maintaining output identity with standard speculative decoding.

Weaknesses:

1. Increased System Complexity. The proposed pre-drafting mechanism, with early exits and client-side caching, adds considerable implementation complexity compared to simpler decoding strategies. The system's performance benefits are also highly dependent on the quality of early exit predictions, edge device latency, and communication latency, which might be hard to predict.

2. Early Exit Training Overhead. The necessity of training early exit adapters introduces an additional, non-trivial step and computational cost (100+ GPU hours per model).

3. Communication overhead. Additional communication rounds are required to facilitate the verification at finer granularity.

---

### Review · Reviewer_n1bg · 2025-06-14

**Summary Of Contributions:**

A novel edge-cloud LLM decoding framework with early exits and preemptive drafting significantly cuts latency (up to 35%) and costs (up to 52%) while preserving accuracy, demonstrating real-time applicability on a robot.

**Audience:**

Yes

**Broader Impact Concerns:**

The paper, "Fast and Cost-effective Speculative Edge-Cloud Decoding with Early Exits," primarily focuses on optimizing LLM deployment. However, it presents two areas of ethical implications that warrant a Broader Impact Statement would be (1) Environmental impact ---  While the paper emphasizes cost-effectiveness and efficiency, which can implicitly reduce energy consumption by optimizing API calls and leveraging edge devices, it does not explicitly quantify the potential environmental benefits or drawbacks of this framework. A broader impact statement could address the energy footprint of running LLMs, especially with the distributed nature of the proposed system, and whether this approach genuinely leads to a net reduction in carbon emissions compared to solely cloud-based solutions. (2) Dependence on Specific Hardware and Models: The evaluation highlights performance on NVIDIA Jetson Nano and A100 GPU with specific Vicuna and Llama models. This reliance on particular hardware and models could create dependencies or exclusionary practices if the framework is not easily adaptable to diverse open-source or more accessible alternatives.

**Claims And Evidence:**

Yes

**Requested Changes:**

Adjustments to Strengthen the Work:

(1) The evaluation uses specific hardware configurations like NVIDIA Jetson Nano and A100 GPU. While this is a good start, discussing the potential performance across a wider range of edge-cloud configurations (e.g., lower-power edge devices, varying network latencies, or different server infrastructures beyond a single high-end GPU) would provide a more comprehensive understanding of the framework's scalability and robustness. This could be framed as future work but with more specific considerations.
(2) The paper notes that the current system supports "single-client interaction with the server" and identifies multi-client concurrency as a natural next step. Given that many real-world LLM deployments involve concurrent requests, outlining the key challenges and conceptual approaches for extending the framework to handle multiple concurrent clients would significantly enhance its practical relevance.

**Strengths And Weaknesses:**

Strengths:

(1) The paper directly tackles the significant issues of high operational costs and latency associated with deploying large language models (LLMs) on edge devices, which is a major barrier for smaller organizations and real-time applications like robotics.
(2) The proposed "speculative edge-cloud decoding framework with early exits" presents a novel solution that effectively leverages both edge and cloud resources. This split between a small draft model on the device and a large target model on the server is a key innovation.
(3) The method demonstrates impressive quantitative improvements, including up to a 35% reduction in latency compared to cloud-based autoregressive decoding, and an additional 11% improvement from preemptive drafting. It also achieves cost reductions of up to 52% over cloud autoregressive decoding.
(4) Real-World Validation: The deployment and testing on a Unitree Go2 quadruped robot for vision-language model (VLM) based control provides strong evidence of the framework's practical applicability and effectiveness in resource-constrained, real-time scenarios.
(5) The authors claim this is the first work to demonstrate end-to-end speculative decoding with models split between edge and cloud, highlighting its pioneering nature.

Weaknesses
(1) While the paper states the method "operates effectively without making assumptions about communication delays"  and that priority queues help optimize bandwidth , it also acknowledges that "communication latency can be a limiting factor in extreme cases". Further elaboration on what constitutes "extreme cases" and how the system's performance degrades under such conditions would be beneficial.
(2) While tested on Jetson Nano and A100 GPU, the performance in other edge-cloud configurations (e.g., different network latencies, lower-power edge devices, or various server infrastructures beyond a single GPU) could be further explored or discussed as a future work direction.
(3) The authors state that the Python implementation "leaves room for further optimization" and suggest a low-level C++ implementation for latency-sensitive applications. While understood for a proof-of-concept, a brief discussion of the expected performance gains from such optimizations could provide more complete context.
(4)  The current system supports "single-client interaction with the server," with multi-client concurrency noted as future work. Given that many real-world LLM deployments involve concurrent requests, elaborating on the challenges and proposed solutions for scaling to multi-client scenarios would be important for broader impact.

---

### Review · Reviewer_zGA5 · 2025-06-26

**Summary Of Contributions:**

This paper presents a novel and practical framework for speculative edge-cloud decoding that enables fast and cost-effective deployment of large language models (LLMs) on resource-constrained edge devices. The key contributions include several aspects: Edge-Cloud Collaboration for Speculative Decoding, Early Exits with Preemptive Drafting, Real-World Deployment on Robotics Platform. Overall, the work introduces a significant advancement toward scalable, real-time LLM inference in edge-cloud scenarios, addressing both latency and cost-efficiency without sacrificing output fidelity.

**Audience:**

Yes

**Broader Impact Concerns:**

This work does not involve any ethical issues.

**Claims And Evidence:**

Yes

**Requested Changes:**

1. Clarify Cache Miss Implications on Low-End Devices: Provide more analysis or discussion on how this affects the overall system performance and whether adaptive exit thresholds or model-specific tuning can help mitigate it.

2. Detail Early Exit Training Procedure and Threshold Selection: Expand on how thresholds are set (fixed, learned, heuristic?) and provide metrics or plots showing the confidence distribution at different layers to justify early exit effectiveness.

**Strengths And Weaknesses:**

Strengths:
1. Novel Edge-Cloud Framework:
The paper introduces a well-motivated and original speculative decoding framework that splits draft and target models across edge and cloud, addressing a key challenge in deploying LLMs on resource-constrained devices.

2. Effective Use of Early Exits and Pre-Drafting:
The incorporation of early exits in the target model to enable client-side pre-drafting is a clever design that improves parallelism and reduces latency without compromising correctness.

3. Comprehensive Evaluation:
The authors provide thorough experiments across multiple model sizes, tasks, and hardware setups (Jetson Nano and RTX), with detailed breakdowns of latency components, cache behavior, and ablation studies. In addition, they implement the method on the Unitree Go2 quadruped robot for vision-language navigation tasks significantly strengthens the paper’s practical relevance.

4. Clear and Structured Presentation:
The paper is well-organized, with clear diagrams and algorithms that enhance understanding of the proposed methodology and system implementation.

Weaknesses:
1. High Cache Miss Rates on Weaker Devices:
The Jetson Nano, a common low-end edge device, exhibits relatively high cache miss rates (up to 70%+), which limits the benefits of pre-drafting in some settings. Further tuning or adaptive mechanisms could be discussed to mitigate this.

2. Limited Multi-Client Support:
The current system is designed for single-client use. Scalability to multi-client scenarios, which is important for real-world applications, is mentioned as future work but not explored experimentally.

3. Early Exit Training Details Could Be Expanded:
While early exit adapters are introduced effectively, the paper could benefit from more discussion on how training stability and exit thresholds impact performance across different models and datasets.

4. Assumption of Stable Connectivity:
The method relies on frequent client-server communication. Although the authors acknowledge bandwidth constraints, the system may struggle in environments with unreliable or low-bandwidth networks, which could be better quantified.

---

> ### Author Response · Authors · 2025-06-30
> **Response to reviewer zGA5**
>
> We thank the reviewer for their thoughtful and encouraging feedback. We have revised the paper to address all of your concerns and include the updated sections here.
>
> **On Cache Miss Rate**: We have expanded our discussion section to address cache miss behavior in detail. Cache miss rates are influenced by both the quality of early exit adapters and the hardware capabilities of the client device. Lower-end devices like the Jetson Nano are constrained in memory and compute, which limits their ability to pre-draft and store tokens for multiple early exit outputs. To mitigate this, we prioritize early exits with higher confidence; however, these often lie deeper in the model, increasing compute demands. We now also highlight that smarter scheduling strategies beyond static confidence prioritization (e.g., delayed or adaptive speculative drafting) could help balance latency and cache efficiency.
>
> **On Early Exit Procedure**: We clarify that our system does not use fixed or learned thresholds to trigger early exits in the Early Exit Training Section (Section 3; last paragraph). Instead, all exits are evaluated, and we prioritize communication and pre-drafting of early exits based on the confidence (softmax probability) of the top token prediction. As shown in Fig. (https://drive.google.com/file/d/1fF-S8GVvHJllSCAAEtMZ7rNRTZgcpEMp/view), confidence scores tend to increase with layer depth, and later exits generally have stronger alignment with the full model’s output. We have included a Section on Early Exit Analysis in the Appendix for further details.
>
> **Communication Latency**: The Discussion section now defines “extreme” connectivity scenarios (e.g., RTT > 300–400 ms or unstable mobile networks) and explains how our system may degrade under such conditions. We describe how our priority queue mechanism is designed to improve bandwidth utilization and mitigate these effects. Future directions include adaptive communication and dynamic scheduling.
>
> **Multi-Client Scalability**: We have elaborated on multi-client support as a natural extension of our system. The Discussion outlines challenges such as queue synchronization, resource contention, and bandwidth fairness and proposes potential solutions, including batched verification and client prioritization.

---

> > ### Author Response · Authors · 2025-06-30
> > **Response to reviewer zGA5**
> >
> > **Updated Early Exit Training Section (Section 3; last paragraph)**: We add adapter layers after each layer of the target model to train the early exits, as shown in Fig. 3. Each adapter connects to the language model (LM) head, and its loss is backpropagated to update only that adapter. This approach minimizes the number of trainable parameters and preserves the original model weights. Importantly, we do not define fixed confidence thresholds to determine when to exit; instead, all exit outputs are computed during inference, and their associated confidence scores are used to prioritize which outputs to communicate first.
> >
> > **Early Exit Analysis (Appendix D)**: In Fig. ( https://drive.google.com/file/d/1fF-S8GVvHJllSCAAEtMZ7rNRTZgcpEMp/view), we present a detailed comparison of token match percentages and average confidence scores across early exit layers in the Vicuna-7B model. The token match percentage quantifies the proportion of tokens generated by each early exit that match the corresponding tokens in the final model output. As expected, this match percentage increases steadily with deeper exits, indicating that later layers are more aligned with the final output distribution. Interestingly, the match percentage grows sharply beyond the halfway point of the network, suggesting that substantial refinement of predictions occurs in the latter half of the model’s depth. Notably, early exits in the initial layers achieve only 20–30\% token-level agreement with the final model, underscoring the limited utility of very shallow exits for high-fidelity generation.
> >
> > Alongside match rates, we also plot the average confidence score at each early exit, defined as the softmax probability of the most likely token at each timestep. While confidence generally correlates with layer depth, we observe that confidence saturates earlier than match accuracy. This suggests that some early exits, particularly in the mid-range layers, can achieve high token match percentages despite lower confidence scores, indicating that accurate predictions may be made even before the model’s certainty peaks. Such behavior can be leveraged for adaptive inference: exits that combine high match rates with sufficient confidence offer promising points for early stopping, enabling latency reduction without substantial loss in output quality. These findings highlight the value of jointly analyzing confidence and output fidelity when designing efficient decoding strategies based on early exits.

---

> > > ### Author Response · Authors · 2025-06-30
> > > **Response to reviewer zGA5**
> > >
> > > **Added to Discussion Section (Section 6)**:
> > >
> > > Cache miss rates significantly affect the efficiency of speculative decoding and are primarily influenced by two factors: (i) the quality of the early exit adapters and (ii) the computational capacity of the client device. Well-trained and calibrated early exits are more likely to produce outputs that match the final model, thereby increasing cache hit rates and improving decoding throughput. However, on resource-constrained devices like the Jetson Nano, limited memory and compute capabilities restrict the number of early exit outputs that can be computed and stored in the pre-draft cache. Although we train early exit adapters on general-purpose conversational data, task-specific fine-tuning can improve their alignment with final outputs, thereby reducing cache misses. On the client side, we prioritize speculative drafts from higher-confidence exits. However, these typically correspond to later layers, which are delayed in arriving to the client. Future extensions could explore more sophisticated scheduling strategies beyond static confidence-based prioritization, potentially introducing controlled delays to improve hit rates while balancing latency and resource usage.

---

### Decision · Action_Editor_izB1 · 2025-08-08

**Recommendation:** Accept as is

**Audience:**

Yes

**Audience Explanation:**

Yes, speculative decoding, inference on edge devices, and early exiting are all of potential interest to some members.

**Claims And Evidence:**

Yes

**Claims Explanation:**

Reviewers pointed out that the paper had comprehensive experiments, covering many different models in real-world setups with extensive breakdowns of performance (including latency, accuracy, etc.). Reviewers also appreciated that the paper includes a case study on a robot.